# Clinical learning environments and experiences of nursing students in West Bank Universities: A mixed-methods study

**Nihad Bsharat[1], Ibrahim Aqtam[1]\*, Hanan Al-Modallal[2], Mutaz Dreidi[3], Mustafa Shouli[1]**

**1** Ibn Sina College for Health Professions, Department of Nursing, Nablus University for Vocational and Technical Education, Nablus, Palestine, **2** The Hashemite University, Faculty of Nursing, Department of Community and Mental Health Nursing, Zarqa, Jordan, **3** Department of Nursing, Faculty of Pharmacy, Nursing and Health Professions, Birzeit University, Ramallah, Palestine

\* ibrahim.aqtam@nu-vte.edu.ps, info@nu-vte.edu.ps

## Abstract

### Background

Clinical learning environments (CLEs) play a vital role in shaping nursing students' competencies, yet their dynamics in conflict-affected settings remain underexplored. This study investigated how CLE factors, measured via the Clinical Learning Environment, Supervision and Nurse Teacher (CLES+T) scale, relate to students' clinical learning (CL) experiences across West Bank universities in Palestine. The findings are specific to the Palestinian context but may offer insights for other conflict-affected educational settings.

### Methods

A convergent mixed-methods design integrated quantitative data from 306 nursing students across governmental, public, and private universities with qualitative insights from 14 in-depth interviews. The validated Arabic CLES+T scale, culturally adapted for the Palestinian context, assessed five CLE dimensions. Quantitative analysis included ANOVA, correlation, and hierarchical multiple regression. Assumptions for parametric tests (normality, multicollinearity, and homoscedasticity) were verified before conducting ANOVA and regression analyses. Qualitative data underwent inductive content analysis with dual coding by two independent researchers to ensure reliability.

### Results

CLE dimensions were strongly correlated with students' clinical learning experiences ($r = 0.758$, $p < 0.001$). Pedagogical atmosphere ($\beta = 0.365$, $p < 0.001$) and supervisory relationships ($\beta = 0.264$, $p = 0.001$) were significant predictors, jointly explaining 59% of the variance ($R^2 = 0.59$). Students reported more positive CL

**Data availability statement:** The datasets generated and analyzed during this study are not publicly available due to ethical restrictions protecting participant confidentiality and the sensitive geopolitical context of the research. The Palestinian setting necessitates additional privacy safeguards because of potential security risks to participants. However, anonymized data supporting the findings may be made available to qualified researchers upon reasonable request, subject to approval by the Arab American University-Palestine Institutional Review Board (IRB). Requests should be directed to the corresponding author (Dr. Ibrahim Aqtam; ibrahim.aqtam@nu-vte.edu.ps) and will require a formal data-sharing agreement outlining permitted uses.

**Funding:** The author(s) received no specific funding for this work.

**Competing interests:** The authors have declared that no competing interests exist.

experiences in governmental hospitals (M = 3.92 ± 0.91) compared to private facilities (M = 3.59 ± 1.06, p = 0.032). Significant differences emerged across clinical wards (F[6,299]=2.56, p = 0.019), with orthopedic wards receiving the highest scores and pediatric wards the lowest. Qualitative findings highlighted four themes: students' perceptions of clinical experiences, key facilitators (e.g., instructor expertise), barriers (e.g., movement restrictions, limited resources), and strategies for improvement (e.g., expanding clinical exposure and diversifying placement sites).

## Conclusion

In the conflict-affected Palestinian context, positive pedagogical atmospheres and supportive supervisory relationships substantially enhance nursing students' clinical learning. These human factors may help offset systemic and geopolitical barriers. Findings support targeted efforts in faculty development, clinical teaching strategies, and institutional coordination to build educational resilience.

## Introduction

Clinical learning environments (CLEs) are fundamental to nursing education, enabling students to translate theoretical knowledge into professional practice. These environments significantly influence the development of clinical competencies, professional identity, and readiness to deliver patient-centered care [1,2]. Central elements of the CLE, such as pedagogical atmosphere, supervisory relationships, and organizational support, are closely linked to students' clinical learning (CL) experiences, shaping their confidence, skill acquisition, and transition into the workforce [3,4].

Despite substantial research on CLEs in high-income and politically stable settings, there is limited understanding of how these environments function in conflict-affected regions. This study addresses three critical research questions: (1) How do CLE components, as measured by the CLES+T scale, influence nursing students' clinical learning experiences in the Palestinian context? (2) What are the specific conflict-related barriers that affect students' access to and engagement in clinical education? (3) What adaptive strategies do students and educators develop to maintain clinical learning quality despite adverse conditions?

The novelty of this research lies in its application of a validated CLE assessment tool within a conflict-affected educational context, providing unprecedented insights into how educational resilience can be fostered in nursing programs operating under geopolitical constraints. This contributes to the limited literature on nursing education in conflict zones and offers evidence-based recommendations for similar contexts globally.

In such contexts, educational systems are often strained by geopolitical instability, inadequate infrastructure, and resource scarcity, presenting distinct challenges to nursing education [5,6]. These circumstances complicate the delivery of consistent and high-quality clinical instruction, which is critical for safe and effective nursing practice.

Recent research on stress and coping behaviors among nursing students in clinical settings has demonstrated the significant impact of environmental factors on student well-being and learning outcomes. Studies by Toqan et al. [7] have shown that progressive muscle relaxation exercises can effectively reduce anxiety among nursing students during their initial clinical training, highlighting the importance of stress management interventions in clinical education [8]. Furthermore, research examining stress sources and coping behaviors among nursing students throughout their first clinical training has identified specific environmental stressors that impede learning, including inadequate supervision, unfamiliar clinical settings, and performance anxiety [9]. Additionally, investigations into the effectiveness of progressive muscle relaxation in pediatric clinical training contexts have demonstrated that targeted interventions can improve student confidence and reduce barriers to learning [7]. Similar findings have been reported in physical therapy education, where stress management and supportive clinical environments have been shown to enhance student performance and satisfaction [10].

In the occupied Palestinian territories, the nursing education system operates within a uniquely challenging context. Movement restrictions imposed by military checkpoints frequently delay or obstruct students' access to clinical sites, leading to missed learning opportunities and increased stress [11]. The number of checkpoints in the West Bank rose by 32% between 2021 and 2023, further complicating academic scheduling and clinical rotations [11]. Overcrowded hospitals, limited faculty availability, and insufficient simulation resources intensify the theory-practice gap [12]. According to the Palestinian Ministry of Health and Higher Education, 8,156 nursing students were enrolled across 14 institutions in 2023, but available training sites are unable to meet this demand adequately [13]. These structural barriers mirror those found in other low-resource settings, where constraints reduce student engagement and compromise clinical performance [14,15].

Yet, few empirical studies have examined how such systemic and contextual pressures intersect with CLE dynamics in Palestine. The current research addresses this gap by using a validated and culturally adapted Arabic version of the Clinical Learning Environment, Supervision and Nurse Teacher (CLES+T) scale [16], which has shown strong psychometric properties in several Arab countries, including Saudi Arabia, Jordan, and Lebanon [17–19].

This study is grounded in Dunn and Burnett's organizational and educational framework [20], which conceptualizes learning environments as products of structural, interpersonal, and pedagogical forces. This theoretical foundation provides a comprehensive understanding of CLE functionality in the Palestinian context, allowing for the identification of modifiable factors that can enhance clinical learning outcomes despite resource and political constraints.

By integrating quantitative and qualitative data, this research identifies modifiable factors that can enhance clinical learning outcomes despite resource and political constraints. The findings aim to inform nursing education policy, curriculum planning, and faculty development not only in Palestine but also in other regions facing similar disruptions.

## Materials and methods

### Study design and setting

A convergent mixed-methods design was employed to examine the clinical learning environments (CLEs) and experiences of undergraduate nursing students across three universities in the West Bank, representing governmental, public, and private institutions. The quantitative component included a cross-sectional survey of 306 nursing students, while the qualitative component comprised semi-structured interviews with 14 purposively selected fourth-year students. Data collection took place from September to December 2023 across various clinical training sites, including governmental hospitals, private clinics, and United Nations Relief and Works Agency (UNRWA) facilities.

### Participants and sampling

**Sampling strategy and power analysis.** A convenience sampling strategy was employed for the quantitative phase, stratified by university type (governmental, public, private) to ensure representation across institutional categories. Based

on official enrollment distributions from the Palestinian Ministry of Higher Education [13], proportional representation was ensured: governmental (6%), public (35%), and private (59%).

A priori power analysis using G*Power 3.1 determined the required sample size. For multiple regression with five predictors, a sample of 138 was needed (medium effect size $f^2 = 0.15$, $\alpha = 0.05$, power = 0.90). The use of power = 0.90 instead of the conventional 0.80 was chosen to increase the statistical power and reduce the risk of Type II error, given the exploratory nature of this study in a conflict-affected context where effect sizes might be more variable and difficult to detect. For ANOVA across three groups, a minimum of 207 participants was required ($f = 0.25$, $\alpha = 0.05$, power = 0.90). The final sample of 306 exceeded both thresholds.

Eligibility criteria included: (1) enrollment in second to fourth year of a bachelor's nursing program, (2) completion of at least one clinical rotation, (3) current participation in clinical training, (4) age range of 19–26 years, and (5) both male and female students were included to ensure gender representation. First-year students were excluded due to a lack of clinical exposure, as were part-time or diploma-to-degree bridge students due to differing clinical pathways.

For the qualitative phase, purposive sampling targeted fourth-year students across all three university types to ensure rich, contextually diverse narratives. These participants were chosen for their cumulative clinical experience across various departments. Saturation, defined as the absence of new codes or themes in three consecutive interviews, determined the final qualitative sample size of 14 students [21].

## Instrument

**Clinical learning environment, supervision, and nurse teacher (CLES+T) scale.** The CLES+T scale, developed by Saarikoski et al. [16], is a 34-item tool assessing five CLE dimensions: pedagogical atmosphere, leadership style of the ward manager, premises of care, supervisory relationship, and role of the nurse teacher. Each item is rated on a 5-point Likert scale from 1 ("fully disagree") to 5 ("fully agree").

For this study, the validated Arabic version of the CLES+T scale [17] was used and culturally adapted to the Palestinian context. Minor terminological adjustments, such as translating "nurse teacher" as "المدرس السريري" [clinical instructor] and "ward" as "قسم", were made to reflect local healthcare terminology. Adaptations were reviewed by a panel of experts (two nursing educators, one clinical instructor, one linguist, and one senior student), achieving a Content Validity Index (CVI) of 0.92, indicating excellent content validity [22]. The adapted scale demonstrated high internal consistency (Cronbach's $\alpha = 0.984$), consistent with previous Arabic validations conducted in Saudi Arabia [17], Jordan [18], and Lebanon [19]. In the current study, the reliability scores (Cronbach's alpha) for the CLES+T scale dimensions were as follows: pedagogical atmosphere ($\alpha = 0.91$), leadership style of ward manager ($\alpha = 0.89$), premises of care ($\alpha = 0.87$), supervisory relationship ($\alpha = 0.93$), and role of nurse teacher ($\alpha = 0.88$), all indicating excellent internal consistency.

## Data collection

**Quantitative data collection.** Paper-based questionnaires were administered during pre-scheduled sessions coordinated with the nursing departments to avoid academic disruption. Research assistants distributed the CLES+T instrument in classroom settings, providing standardized instructions and clarification without influencing responses. Students completed the surveys in approximately 25–30 minutes. A total of 306 valid responses were collected, yielding a 92.7% response rate.

**Qualitative data collection.** Semi-structured interviews were conducted with 14 fourth-year students (5 males, 9 females), representing all university types. Interviews lasted 45–60 minutes and were held in private rooms at the participants' universities to ensure comfort and confidentiality. An interview guide was developed to explore perceptions of the clinical environment, facilitators and barriers to learning, and suggested improvements. All interviews were audio-recorded with consent and transcribed verbatim. Field notes captured non-verbal cues and contextual observations. The coding procedures involved initial open coding of transcripts, followed by axial coding to identify relationships between

codes, and selective coding to develop main themes. Two independent researchers conducted the coding process, with discrepancies resolved through discussion and consensus.

### Data analysis

**Quantitative analysis.** Statistical analyses were performed using SPSS version 26. Descriptive statistics (frequencies, percentages, means, standard deviations) described the sample and CLES+T scores. Prior to conducting parametric tests, the following assumptions were verified: (1) normality was assessed using the Kolmogorov-Smirnov test and visual inspection of Q-Q plots, (2) multicollinearity was evaluated using Variance Inflation Factor (VIF) values, with all values below 4.0 indicating acceptable limits, and (3) homoscedasticity was confirmed through Levene's test and residual plots. Pearson's correlation coefficients examined relationships between CLE dimensions and clinical learning experiences. One-way ANOVA and Tukey's post-hoc tests compared outcomes across university types, clinical sites, and ward placements. Hierarchical multiple regression was used to identify predictors of clinical learning experiences, controlling demographic variables.

**Qualitative analysis.** Inductive content analysis followed Elo and Kyngäs's methodology [23]. Transcripts were openly coded, and similar codes grouped into subcategories and abstracted into main themes. Two researchers independently coded the data, with inter-coder reliability assessed through Cohen's kappa coefficient ($\kappa = 0.85$), indicating strong agreement. Discrepancies were resolved through discussion. Member checking was conducted with five participants to validate the thematic interpretations. NVivo 12 software supported data management and coding.

### Integration of quantitative and qualitative findings

Following mixed-methods research guidelines [24], a joint display approach was used to integrate findings. At the methods level, interview questions elaborated on themes assessed by the CLES+T scale. At the interpretation level, triangulation of quantitative results and qualitative insights revealed converging, complementary, and contrasting findings. These were presented in an integrated joint display table (Table 3). Methodological transparency was enhanced through the completion of a Checklist of Mixed Methods Elements [25] (Supplementary File 1–4).

### Ethics approval and consent to participate

Ethical approval was obtained from the Arab American University-Palestine Institutional Review Board (IRB Approval No: AAUP/IRB/2023-012). Formal permissions to access students and clinical training sites were granted by the university presidencies and deans of nursing faculties at all participating institutions. All participants received written informed consent forms detailing the study purpose, procedures, confidentiality safeguards, and voluntary nature of participation. Students were explicitly informed that their participation was entirely voluntary, that they could withdraw at any time without academic penalty, and that their responses would remain confidential and anonymous. To ensure voluntariness and avoid coercion, data collection sessions were conducted independently of faculty evaluation periods, and instructors were not present during questionnaire administration.

Data security measures included: (1) all physical questionnaires were stored in locked cabinets accessible only to the research team, (2) digital data files were encrypted and password-protected, (3) participant identifiers were separated from response data, and (4) all data will be destroyed after the mandatory 5-year retention period.

All procedures complied with the ethical standards outlined in the Declaration of Helsinki.

## Results

### Participant characteristics

The quantitative sample (N = 306) included nursing students from private (59.5%, n = 182), public (34.9%, n = 107), and governmental (5.6%, n = 17) universities. The majority were female (68.3%, n = 209) with a mean age of 21.4 years

(SD = 1.8). Most participants were in their fourth year (42.8%), followed by third year (32.7%) and second year (24.5%). Clinical training occurred across governmental hospitals (45.8%), private hospitals (38.2%), and UNRWA clinics (16.0%). Common ward placements included medical (28.8%), surgical (22.5%), and intensive care units (15.7%).

The qualitative sample (n = 14) included nine females and five males, all in their final year of study. Participants represented all three university sectors (private = 8, public = 4, governmental = 2) and reported a mean of 6.3 distinct clinical placements (range: 4–9).

## Quantitative findings

**Clinical learning environment (CLES+T) dimensions.** Table 1 presents descriptive statistics for the five dimensions of the CLES+T scale. Among the subscales, supervisory relationship was rated highest (M = 3.78, SD = 1.02), followed by pedagogical atmosphere (M = 3.65, SD = 0.98). The leadership style of the ward manager received the lowest average score (M = 3.24, SD = 1.15). The overall CLES+T mean score was 3.52 (SD = 0.97), indicating moderate-to-high perceptions of the clinical learning environment.

Bivariate correlation analysis indicated strong positive relationships between all CLE dimensions and students' reported clinical learning experiences (r = 0.758, p < 0.001). The pedagogical atmosphere dimension had the strongest individual correlation with learning experiences (r = 0.741, p < 0.001), followed by supervisory relationships (r = 0.726, p < 0.001).

## Predictors of clinical learning experiences

Hierarchical multiple regression analysis was conducted to examine the predictive power of the CLES+T subscales on students' clinical learning experiences (Table 2). The sample size calculation was based on the rule of having at least 15 cases per predictor variable, and with 12 predictors in the final model, the minimum required sample was 180 participants. Our sample of 306 exceeded this requirement, ensuring adequate statistical power. In the first step, demographic variables accounted for 8% of the variance (R² = 0.08, p < 0.001). When the five CLES+T dimensions were added in the second step, the model explained an additional 51% of the variance (ΔR² = 0.51, p < 0.001), yielding a total $R^2$ of 0.59 (F[12,293] = 35.27, p < 0.001). Among the predictors, the pedagogical atmosphere (β = 0.365, p < 0.001) and supervisory relationship (β = 0.264, p = 0.001) emerged as statistically significant, while the remaining dimensions, leadership style of the ward manager, premises of care, and role of the nurse teacher, did not reach significance. Variance Inflation Factor (VIF) values ranged from 1.12 to 3.90, indicating no substantial multicollinearity among the predictors.

## Differences across clinical training contexts

One-way ANOVA revealed significant differences in learning experiences across training sites (F[2,303] = 3.47, p = 0.032, η² = 0.023). Post-hoc analysis showed that students placed in governmental hospitals reported significantly higher learning experiences (M = 3.92, SD = 0.91) compared to those in private hospitals (M = 3.59, SD = 1.06, p = 0.032). UNRWA facilities received relatively high ratings (M = 3.87, SD = 0.88) but were not statistically different from the others.

**Table 1. Mean Scores of Clinical Learning Environment Dimensions (N = 306).**

| CLES+T Dimension | Mean | SD | Range |
|---|---|---|---|
| Supervisory relationship | 3.78 | 1.02 | 1-5 |
| Pedagogical atmosphere | 3.65 | 0.98 | 1-5 |
| Role of nurse teacher | 3.52 | 1.08 | 1-5 |
| Premises of care | 3.41 | 1.12 | 1-5 |
| Leadership style of ward manager | 3.24 | 1.15 | 1-5 |
| **Overall CLES+T score** | 3.52 | 0.97 | 1-5 |

The 'Role of nurse teacher' dimension was linguistically adapted as 'clinical instructor' (المدرس السريري) in the Arabic instrument.

**Table 2. Hierarchical Multiple Regression Predicting Clinical Learning Experiences (N = 306).**

| Variable | Model 1 β | p | Model 2 β | p |
|---|---|---|---|---|
| **Step 1: Demographics** | | | | |
| Age | 0.112 | 0.068 | 0.043 | 0.312 |
| Gender (female) | 0.156 | 0.007** | 0.072 | 0.084 |
| Academic year | 0.143 | 0.019* | 0.065 | 0.127 |
| University type | | | | |
| Public vs. Private | 0.084 | 0.178 | 0.037 | 0.391 |
| Governmental vs. Private | 0.097 | 0.112 | 0.042 | 0.324 |
| Training site | | | | |
| Governmental vs. Private | 0.165 | 0.006** | 0.076 | 0.079 |
| UNRWA vs. Private | 0.138 | 0.023* | 0.061 | 0.146 |
| **Step 2: CLE Dimensions** | | | | |
| Pedagogical atmosphere | | | 0.365 | <0.001*** |
| Leadership style of ward manager | | | 0.087 | 0.281 |
| Premises of care | | | 0.092 | 0.247 |
| Supervisory relationship | | | 0.264 | 0.001** |
| Role of nurse teacher | | | 0.103 | 0.192 |
| **Model statistics** | | | | |
| R² | 0.08 | | 0.59 | |
| Adjusted R² | 0.06 | | 0.57 | |
| F | 3.64** | | 35.27*** | |

*Note: β = standardized coefficient; VIF = 1.12-3.90; *p < 0.05, **p < 0.01, **p < 0.001.

Significant differences were also found across clinical wards (F[6,299] = 2.56, p = 0.019, η² = 0.049). Orthopedic wards received the highest scores (M = 4.92, SD = 0.15), while pediatric wards had the lowest (M = 2.97, SD = 1.32). No significant differences were observed across university types (F[2,303] = 1.84, p = 0.161).

## Qualitative findings

Four overarching themes emerged from the inductive content analysis:

**1. Perceptions of clinical experiences.** Students described clinical placements as both challenging and rewarding. Many emphasized the dissonance between classroom learning and the realities of practice:

*"What we learn in classrooms is the ideal scenario, but in hospitals, we face the real world with all its constraints."* (Female, private university)

Real-life patient interactions were consistently described as the most valuable component of learning.

**2. Facilitators of clinical learning.** Instructor expertise and collaborative staff emerged as key facilitators:

*"A good instructor makes all the difference... I feel safe to practice and learn from mistakes."* (Female, governmental university)

Supportive staff who involved students in patient care fostered confidence and learning.

**3. Barriers to clinical learning.** Major barriers included resource limitations, movement restrictions, and non-educational task assignments:

*"We often have 8-10 students sharing one patient..."* (Female, public university)

*"Checkpoints delay us and disrupt the learning day entirely."* (Male, private university)

**4. Strategies for improvement.** Students suggested expanding clinical hours, diversifying placement sites, and improving theory-practice alignment

*"More clinical days and rotations through different facilities would help us understand the healthcare system better."* (Female, private university)

**Joint display: Integrated quantitative and qualitative results**

Table 3 illustrates how qualitative themes aligned with or explained the quantitative findings.

## Discussion

This mixed-methods study explored the clinical learning environments and experiences of nursing students in West Bank universities, shedding light on how pedagogical and supervisory factors interact with contextual barriers in a conflict-affected setting. The findings underscore the critical role of the clinical learning environment (CLE), particularly human and organizational components, in shaping educational outcomes under geopolitical constraints.

The strong correlation between CLES+T dimensions and students' clinical learning experiences ($r = 0.758$) confirms global literature highlighting the centrality of CLE quality in nursing education [1,2]. Notably, pedagogical atmosphere and supervisory relationships emerged as the most significant predictors, jointly accounting for 59% of the variance in learning experiences. While similar patterns have been documented in stable and well-resourced contexts [26,27], the present findings carry added importance in Palestine, where such interpersonal supports may buffer students against chronic resource shortages and mobility restrictions. The modest effect sizes observed for training site ($\eta^2 = 0.023$) and ward type ($\eta^2 = 0.049$) suggest that even small environmental differences can meaningfully shape learning, making these dimensions important intervention points for educators and administrators.

Interestingly, students rated governmental hospitals more positively than private institutions, despite the former's known limitations in staffing and infrastructure. Qualitative findings suggest this may reflect clearer teaching protocols, longer-standing affiliations with nursing schools, and a stronger culture of mentorship among staff in the public sector. In contrast, private facilities may prioritize patient satisfaction and operational efficiency over student learning, sometimes limiting students' hands-on engagement. These insights underscore that organizational culture and instructional intent, more than material resources, can significantly influence perceived learning quality.

**Table 3.** *Integration of Quantitative and Qualitative Results.*

| Quantitative Finding | Qualitative Insight | Integration |
|---|---|---|
| Supervisory relationships predict learning ($\beta = 0.264$, $p = 0.001$) | "A good instructor makes all the difference..." | **Convergence** |
| Pedagogical atmosphere is strongest predictor ($\beta = 0.365$, $p < 0.001$) | "Staff treated us as colleagues-in-training..." | **Convergence** |
| Governmental hospitals rated higher than private ($p = 0.032$) | "Private hospitals restrict what students can do..." | **Complementarity** |
| Orthopedic wards rated highest; pediatric lowest | "Pediatrics limits student practice due to parental presence." | **Complementarity** |
| No difference across university types ($p = 0.161$) | "The university matters less than the clinical site." | **Convergence** |

A similar interpretation applies to UNRWA clinics, which received favorable ratings likely due to their standardized educational protocols and support structures. Operating under international oversight, these clinics often include dedicated staff responsible for guiding and supervising students, demonstrating that institutional consistency and supervision can create meaningful learning environments even in resource-constrained systems.

The qualitative data also emphasized how students navigate the theory-practice gap, a common global challenge [28], but one especially pronounced in Palestine due to movement restrictions, unpredictable clinical access, and political instability. These barriers not only delay physical access to clinical sites but also induce chronic psychological stress, reducing students' ability to focus and retain clinical competencies. A recurrent theme in student narratives was the assignment of non-educational tasks, such as routine patient transport or administrative work, which, while part of basic nursing duties, often crowded out opportunities for skill development. This points to the need for clearer role definitions and institutional agreements that protect students' learning time.

## Limitations

Several limitations should be acknowledged. The cross-sectional design limits causal inferences about the relationships between CLE factors and learning outcomes. Self-report bias may have influenced responses, as students might have provided socially desirable answers about their learning experiences or clinical environments. The underrepresentation of governmental university students (5.6%) mirrors actual enrollment figures but limits robust comparison across institutional types.

Potential selection bias may have occurred despite efforts to ensure representativeness, as students who chose to participate might differ systematically from non-participants. The study was conducted during a relatively stable period in the West Bank, and findings may not generalize to periods of acute conflict or escalated restrictions. Additionally, the convenience sampling approach, while stratified by university type, may limit the generalizability of findings to the broader population of nursing students in Palestine.

Several actionable strategies emerged from the data. First, faculty development in context-sensitive clinical instruction methods can enhance the pedagogical atmosphere, which was the strongest predictor of learning. Even short, structured training for instructors has shown promise in low-resource contexts [29]. Second, formal mentorship programs could strengthen supervisory relationships, fostering trust and learning confidence. Third, innovative educational strategies, such as mobile simulation units, asynchronous virtual case-based learning, and adaptive scheduling to accommodate checkpoint delays, may help overcome logistical barriers. Simulation, in particular, has been shown to effectively substitute or supplement clinical exposure in conflict-affected regions [30].

### Practical implications

This study offers several concrete recommendations for policy-makers, educators, and clinical supervisors.

### For policy-makers

It is essential to develop standardized clinical education protocols across all healthcare facilities to ensure consistency in student learning experiences. Additionally, establishing dedicated funding for clinical instructor training programs focused on conflict-sensitive pedagogy can enhance the quality of instruction. Creating flexible academic calendars that accommodate movement restrictions and security-related disruptions is also crucial to support uninterrupted education. Furthermore, implementing inter-institutional agreements that facilitate student mobility across clinical sites will promote equitable access to diverse clinical learning opportunities.

### For educators

To further strengthen clinical education, it is important to integrate stress management and resilience-building components into clinical preparation courses, equipping students to cope with the demands of healthcare environments. Developing

context-appropriate simulation scenarios that reflect local healthcare challenges ensures that training remains relevant and practical. Establishing peer mentorship programs that pair senior students with junior colleagues can foster a supportive learning culture and enhance knowledge transfer. Additionally, creating mobile clinical education units that can reach students when traditional sites are inaccessible helps maintain continuity in training during times of disruption.

### For clinical supervisors

Enhancing the clinical learning environment also requires implementing structured orientation programs for new clinical instructors, with an emphasis on supportive supervision techniques to foster effective teaching relationships. Developing clear role definitions that protect students' learning time from non-educational tasks is essential to ensure that clinical placements remain focused on education. Establishing regular feedback mechanisms between students and supervisors allows for prompt identification and resolution of concerns, contributing to a more responsive and supportive environment. Finally, creating collaborative learning opportunities that involve students as active participants in patient care promotes deeper engagement and practical skill development.

### Future research directions

Future research should pursue several key directions to deepen understanding and strengthen clinical education in conflict-affected settings. First, longitudinal studies are needed to track nursing students' clinical learning experiences over time, especially during periods of varying conflict intensity, to assess how acute political events impact educational outcomes. Comparative studies examining clinical learning environments in other conflict-affected regions, such as Syria, Yemen, and Afghanistan, can help distinguish between universal and context-specific factors influencing educational resilience. Intervention studies should focus on developing and testing targeted solutions, such as mobile simulation programs, digital mentorship platforms, and conflict-sensitive pedagogical training, to evaluate their effectiveness in enhancing clinical learning outcomes. Additionally, investigating the long-term outcomes of clinical education in these settings, such as professional competence, job satisfaction, and retention in the healthcare workforce, will provide insight into the broader implications of educational disruption. Finally, exploring the integration of technologies like virtual reality, augmented reality, and other digital tools can offer alternative pathways for clinical skill development when traditional access to clinical sites is constrained.

## Conclusion

Despite ongoing systemic challenges in Palestine, this study demonstrates that positive pedagogical environments and effective clinical supervision are strongly associated with improved learning experiences among nursing students. These human factors, embedded in relationships, feedback, and instructional culture, can serve as stabilizing forces that help students cope with and learn within structurally limited and politically volatile settings.

To build educational resilience in conflict-affected regions, targeted interventions should prioritize faculty development, strengthening supervisory roles, and innovative teaching models that adapt to local constraints. Policies must move beyond infrastructure investment alone and address the relational and cultural dimensions of learning, which appear to exert the greatest influence on student outcomes in adversity. Future research should track nursing education outcomes longitudinally, explore conflict-period variations, and examine how clinical learning translates into real-world nursing performance in fragile healthcare systems. Generating this knowledge is essential not only for academic institutions but also for health workforce sustainability in regions burdened by protracted crises.

## Supporting information

**S1 File. Supplementary File 1.**
(DOCX)

**S2 File. Supplementary File 2.**
(DOCX)

**S3 File. Supplementary File 3.**
(DOCX)

**S4 File. Supplementary File 4.**
(DOCX)

## Author contributions

**Conceptualization:** Nihad Bsharat, Ibrahim Aqtam.

**Data curation:** Nihad Bsharat.

**Formal analysis:** Nihad Bsharat, Ibrahim Aqtam, Mutaz Dreidi.

**Investigation:** Nihad Bsharat, Mustafa Shouli.

**Methodology:** Nihad Bsharat, Ibrahim Aqtam, Hanan Al-Modallal.

**Project administration:** Ibrahim Aqtam, Mustafa Shouli.

**Supervision:** Ibrahim Aqtam, Hanan Al-Modallal.

**Validation:** Mutaz Dreidi.

**Writing – original draft:** Nihad Bsharat, Ibrahim Aqtam.

**Writing – review & editing:** Hanan Al-Modallal, Mutaz Dreidi, Mustafa Shouli.

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
