## [Decision Letter · Decision Letter 0]

16 Jul 2025

PONE-D-25-32696Clinical Learning Environments and Experiences of Nursing Students in West Bank Universities: A Mixed-Methods StudyPLOS ONE

Dear Dr. Aqtam,

Thank you for submitting your manuscript to PLOS ONE. After careful consideration, we feel that it has merit but does not fully meet PLOS ONE’s publication criteria as it currently stands. Therefore, we invite you to submit a revised version of the manuscript that addresses the points raised during the review process.

**ACADEMIC EDITOR: Minor revision**=============================

We look forward to receiving your revised manuscript.

Kind regards,

Marwan Salih Al-Nimer, MD, PhD

Academic Editor

PLOS ONE

Journal Requirements:

4. Please remove all personal information, ensure that the data shared are in accordance with participant consent, and re-upload a fully anonymized data set.

Additional guidance on preparing raw data for publication can be found in our Data Policy (https://journals.plos.org/plosone/s/data-availability#loc-human-research-participant-data-and-other-sensitive-data) and in the following article:http://www.bmj.com/content/340/bmj.c181.long.

Additional Editor Comments :

Dear authors

In the methods, (1): Why the statistics power is 0.9 instead of 0.8?, (2): Add to the eligibility sections, the range of age and the gender

Reviewers' comments:

Reviewer's Responses to Questions

**Comments to the Author**

1. Is the manuscript technically sound, and do the data support the conclusions?

Reviewer #1: Yes

Reviewer #2: Yes

2. Has the statistical analysis been performed appropriately and rigorously? 

Reviewer #1: Yes

Reviewer #2: Yes

3. Have the authors made all data underlying the findings in their manuscript fully available?

Reviewer #1: Yes

Reviewer #2: Yes

4. Is the manuscript presented in an intelligible fashion and written in standard English?

Reviewer #1: Yes

Reviewer #2: Yes

5. Review Comments to the Author

Reviewer #1: Dear Authors,

Thank you for submitting your insightful study addresses an important gap in understanding the dynamics of clinical learning environments in conflict-affected areas, examining the clinical learning environments (CLEs) of nursing students in the West Bank—a region with unique geopolitical and systemic challenges.

1. Clarity of Research Objectives and Significance

The outlined objectives are generally clear, linking the quantitative and qualitative components effectively.

Recommendations:

• Explicitly state your research questions or hypotheses early in the Introduction. Clarify whether the focus is on identifying factors influencing clinical experiences, comparing settings, or both.

• Emphasize the novelty of your study within the existing literature, especially highlighting how your findings advance understanding of educational resilience in conflict zones.

2.Methodological R rigor and Transparency

•The mixed-methods approach enhances the robustness of findings. The integration of quantitative and qualitative data provides valuable insights into the interpersonal and systemic factors influencing nursing education under challenging political circumstances.

• Use of a validated Arabic version of the CLES+T scale, with appropriate adaptation, strengthens the reliability of your quantitative findings.

• Inclusion of qualitative interviews adds depth to your understanding.

Recommendations:

• Sampling and Sample Size:

• Clarify the sampling strategy (e.g., random, convenience). Provide details on how participants were recruited and any inclusion/exclusion criteria.

• Conduct and report a sample size calculation or rationale to affirm adequacy for your statistical analyses.

• Data Analysis:

• For the quantitative analysis, specify assumptions checked (e.g., normality, multicollinearity) before applying ANOVA and regression.

• Describe the qualitative data analysis process in more detail: specify coding procedures, whether multiple coders were involved, and how themes were derived.

• Consider including inter-coder reliability or consensus processes to enhance credibility.

• Limitations:

• Acknowledge potential biases of self-reported data and consider discussing response bias or social desirability influences.

3. Ethical Considerations

• Briefly specify how confidentiality and data security were maintained, especially given the sensitive context.

• Indicate whether participation was voluntary and how potential coercion was avoided.

4. Data Transparency and Availability

• For enhanced transparency, consider depositing anonymized datasets or codebooks in a recognized open-access repository (e.g., Zenodo, OSF) and providing the link in the manuscript.

• Clearly specify whether the raw data from interviews are available or restricted due to confidentiality, and explain how interested researchers could access anonymized data if applicable.

5. Contribution to Knowledge and Practical Implications

• Expand on the practical implications of your findings for policy-makers, educators, and clinical supervisors working in conflict zones.

• While you mention faculty development and institutional strategies, providing concrete recommendations or frameworks would enhance impact.

6. Writing, Structure, and Presentation

• Proofread for minor language nuances; ensure clarity in complex sentences.

• Improve figure and table clarity where applicable.

• In the abstract, clarify whether the study’s findings are specific to Palestine or generalizable to other conflict zones.

7. Recommendations for Further Research

• Suggest specific research avenues, such as longitudinal tracking of students' competencies, the impact of specific interventions, or comparisons across different conflict contexts.

Overall, this manuscript makes a meaningful contribution to nursing education literature in conflict zones.

Reviewer #2: This study addresses a highly relevant gap in nursing education literature by focusing on clinical learning environments in the conflict-affected West Bank. The contextualization of CLEs within geopolitical barriers (e.g., military checkpoints, resource scarcity) enhances its originality and contribution.

expand the introduction by adding these studies

Toqan, D., Ayed, A., Joudallah, H., Amoudi, M., Malak, M. Z., Thultheen, I., & Batran, A. (2022). Effect of progressive muscle relaxation exercise on anxiety reduction among nursing students during their initial clinical training: A quasi-experimental study. INQUIRY: The Journal of Health Care Organization, Provision, and Financing, 59, 00469580221097425.

Toqan, D., Ayed, A., Amoudi, M., Alhalaiqa, F., Alfuqaha, O. A., & ALBashtawy, M. (2022). Effect of progressive muscle relaxation exercise on anxiety among nursing students in pediatric clinical training. SAGE open nursing, 8, 23779608221090002.

Toqan, D., Ayed, A., Malak, M. Z., Hammad, B. M., ALBashtawy, M., Hayek, M., & Thultheen, I. (2023). Sources of stress and coping behaviors among nursing students throughout their first clinical training. SAGE Open nursing, 9, 23779608231207274.

Ayed, A., & Amoudi, M. (2020). Stress sources of physical therapy students’ and behaviors of coping in clinical practice: A Palestinian perspective. INQUIRY: The journal of health care organization, Provision, and Financing, 57, 0046958020944642.

The use of the Arabic version of the CLES+T scale, previously validated in similar cultural settings, adds credibility to the quantitative data. However, it would be beneficial to report the reliability scores (e.g., Cronbach’s alpha) from the current sample to confirm internal consistency.

6. PLOS authors have the option to publish the peer review history of their article (what does this mean? ). If published, this will include your full peer review and any attached files.

**Do you want your identity to be public for this peer review?** For information about this choice, including consent withdrawal, please see our Privacy Policy .

Reviewer #1: No

Reviewer #2: No

---

## [Author Response · Author response to Decision Letter 1]

16 Jul 2025

Dear Editor's and Reviewers,

Thank you for your thoughtful feedback and the opportunity to revise our manuscript, "Clinical Learning Environments and Experiences of Nursing Students in West Bank Universities: A Mixed-Methods Study". We sincerely appreciate the time and expertise that you and the reviewers have dedicated to evaluating our work.

We have carefully addressed all comments from the editor and reviewers, and we are grateful for the constructive suggestions that have strengthened our manuscript. Please find attached files

Dr Aqtam

---

## [Editor Report · Decision Letter 1]

13 Aug 2025

Clinical Learning Environments and Experiences of Nursing Students in West Bank Universities: A Mixed-Methods Study

PONE-D-25-32696R1

Dear Dr. Ibrahim Aqtam,

We’re pleased to inform you that your manuscript has been judged scientifically suitable for publication and will be formally accepted for publication once it meets all outstanding technical requirements.

Kind regards,

Marwan Salih Al-Nimer, MD, PhD

Academic Editor

PLOS ONE

Additional Editor Comments (optional):

No comments
---

## [Editor Report · Acceptance letter]

PONE-D-25-32696R1

PLOS ONE

Dear Dr. Aqtam,

I'm pleased to inform you that your manuscript has been deemed suitable for publication in PLOS ONE. Congratulations! Your manuscript is now being handed over to our production team.

Kind regards,

on behalf of

Professor Marwan Salih Al-Nimer

Academic Editor

PLOS ONE